# Chronic and Postprandial Metabolic Responses to a Ketogenic Diet Compared to High-Carbohydrate and Habitual Diets in Trained Competitive Cyclists and Triathletes: A Randomized Crossover Trial

**DOI:** 10.3390/ijerph20021110

**Published:** 2023-01-08

**Authors:** Austin J. Graybeal, Andreas Kreutzer, Kamiah Moss, Petra Rack, Garrett Augsburger, Robyn Braun-Trocchio, Jada L. Willis, Meena Shah

**Affiliations:** 1School of Kinesiology & Nutrition, College of Education and Human Sciences, University of Southern Mississippi, Hattiesburg, MS 39406, USA; 2Department of Kinesiology, Harris College of Nursing and Health Sciences, Texas Christian University, Fort Worth, TX 76129, USA; 3School of Health Promotion and Kinesiology, College of Health Sciences, Texas Woman’s University, Denton, TX 76209, USA; 4Physical Medicine and Rehabilitation, Baylor Institute for Rehabilitation, Dallas, TX 75246, USA; 5Department of Nutritional Sciences, College of Science & Engineering, Texas Christian University, Fort Worth, TX 76129, USA

**Keywords:** ketogenic diet, high-carbohydrate diet, endurance athletes, low energy availability, endurance performance, metabolism, energy expenditure, fat oxidation, carbohydrate oxidation

## Abstract

Extreme carbohydrate deficits during a ketogenic diet (KD) may result in metabolic adaptations reflective of low energy availability; however, the manifestation of these adaptations outside of exercise have yet to be elucidated in cyclists and triathletes. The purpose of this study is to investigate the chronic and postprandial metabolic responses to a KD compared to a high-carbohydrate diet (HCD) and habitual diet (HD) in trained competitive cyclists and triathletes. For this randomized crossover trial, six trained competitive cyclist and triathletes (F: 4, M: 2) followed an ad libitum KD and HCD for 14 d each after their HD. Fasting energy expenditure (EE), respiratory exchange ratio (RER), and fat and carbohydrate oxidation (FatOx and CarbOx, respectively) were collected during their HD and after 14 d on each randomly assigned KD and HCD. Postprandial measurements were collected on day 14 of each diet following the ingestion of a corresponding test meal. There were no significant differences in fasting EE, RER, FatOx, or CarbOx among diet conditions (all *p* > 0.050). Although postprandial RER and CarbOx were consistently lower following the KD meal, there were no differences in peak postprandial RER (*p* = 0.452), RER incremental area under the curve (iAUC; *p* = 0.416) postprandial FatOx (*p* = 0.122), peak FatOx (*p* = 0.381), or FatOx iAUC (*p* = 0.164) between the KD and HD meals. An ad libitum KD does not significantly alter chronic EE or substrate utilization compared to a HCD or HD; postprandial FatOx appears similar between a KD and HD; this is potentially due to the high metabolic flexibility of cyclists and triathletes and the metabolic adaptations made to habitual high-fat Western diets in practice. Cyclists and triathletes should consider these metabolic similarities prior to a KD given the potential health and performance impairments from severe carbohydrate restriction.

## 1. Introduction

It is well known that endurance athletes are continually seeking new dietary strategies to improve performance during periods of intensive training. While several professional organizations recommend a high-carbohydrate diet (HCD) [1] for endurance athletes, reports suggest that the dietary habits of endurance athletes are misaligned with these recommendations in practice [2,3], particularly for female endurance athletes who tend to restrict carbohydrates to a greater degree [4]. Given the competitive nature of endurance athletes and the inherent difficulty of consuming carbohydrates at the amounts necessary for performance improvements, it is common for endurance athletes to attempt alternate dietary strategies to gain a competitive advantage. Many of these alternate strategies, however, focus on carbohydrate restriction, with the most popular being the ketogenic diet (KD).

The KD, defined as restricting carbohydrates to <10% of daily energy [5], is used to achieve ketosis, the metabolic state in which the body increases ketone body production and relies on fat as its primary energy source [6]. Specifically, endurance athletes employ the KD to shift the body toward greater fat utilization, otherwise known as “fat adaptation”, in situations where the body would typically and predominately rely on glycogen utilization [7]. The fat-adapted state can occur within five-to-six days on the KD [8,9] and is thought to be associated with increased improved performance and glycogen sparing [6,10,11]. Additionally, it is common for endurance athletes to follow a KD to promote weight loss and improve performance through increases in speed or power.

Despite the potential performance benefits that may result from a KD, endurance athletes rely on a delicate balance between training and dietary intake to avoid physical distress. However, maintaining sufficient energy intake during periods of progressive training is fundamentally difficult, and intentional deficits in carbohydrate intake without corresponding changes in training intensity may result in low energy availability (LEA) and eventual physiological impairment. The health and performance impairments associated with LEA are often reflected in lower energy expenditure (EE) and resting metabolic rate (RMR) [12,13], where these metabolic responses routinely adapt to changes in exercise and dietary intake to conserve energy [14,15,16,17].

While the metabolic maladaptations that occur during rigid dietary interventions have been well-documented in endurance athletes, these findings are often a product of lower energy intake rather than substrate availability, although recent reports have shown that extreme carbohydrate restriction leads to symptoms of relative energy deficiency in sport (RED-S) independent of energy availability [18]. While it is possible that the KD may result in metabolic adaptations reflective of LEA, many studies report in opposition, showing increases in EE at rest [19] and during exercise [20] following a KD in addition to a greater fat utilization during training [7]. However, the majority of the existing literature has determined the metabolic responses to a KD: (1) in individuals with overweight or obesity who may not have the metabolic flexibility of trained endurance athletes; (2) in comparison to a HCD; or (3) during exercise rather than after a meal corresponding to a KD. Considering that cyclists and triathletes follow dietary patterns more akin to a high-fat Western diet, have a high degree of metabolic flexibility, and are in the interprandial state more often than training, it is important to determine the chronic and postprandial metabolic responses to a KD in relation to typical or recommended dietary patterns. However, this has yet to be elucidated in cyclists and triathletes. Therefore, the purpose of this study is to investigate chronic and postprandial EE, respiratory exchange ratio (RER), and fat and carbohydrate oxidation (FatOx and CarbOx, respectively) following an ad libitum KD compared to an ad libitum HCD and habitual diet (HD) in trained competitive cyclists and triathletes.

## 2. Materials and Methods

The current study is a secondary analysis that stems from a larger line of research. The primary outcome variables for this study are independent of the outcome variables included in the larger body of work and only the information describing the sample and control variables are shared. Furthermore, the methods for conducting the current study have been published previously and, thus, are summarized below [21]. A detailed description of the methods can be found elsewhere [21], but any critical additions or differences to the methods specific to this study are described in detail. The pre-registration for this study can be found at ClinicalTrials.gov Identifier: NCT04097171 (accessed 5 October 2021).

### 2.1. Participants

Six trained competitive cyclist and triathletes (F: 4, M: 2) between the ages of 18 and 65 (37.2 ± 12.2 years) completed this randomized crossover trial in its entirety. Participants were considered to be trained competitive based on previous guidelines [22], which included cycling ≥100 km weekly during the past year and having a V̇O2max at or above the 80th percentile for their age and sex (46.6 ± 6.7 mL/kg/min). Participants who did not meet these criteria, were trying to lose weight, or were following a KD or HCD were excluded. A complete list of the exclusion criteria are presented in previous works [21]. The study was approved by the Texas Christian University Institutional Review Board (IRB #192-14) and conducted according to the guidelines put forth in the Declaration of Helsinki. Written informed consent was obtained from all participants prior to participation.

### 2.2. Design

For this randomized crossover trial, participants were tested at baseline during their HD and were subsequently assigned to both a 14-day KD and HCD in a random order. After 14 d of their first randomly assigned diet, participants were immediately crossed over to the other treatment. Since it takes approximately three-to-five days to reach ketosis during a KD [23] and because ketosis is immediately washed out after initiating a HCD, a defined washout period was not used. Similarly, and because of the polarity between the dietary interventions, the effects of HCD are quickly washed out during severe carbohydrate restriction from a KD, which is supported by our findings of either capillary or urinary βHB levels ≥0.5 mmol/L by day 7 for both participants assigned to the HCD before the KD. Fasting and postprandial respiratory gases were collected after each condition (including the HD). The test meals used to measure postprandial metabolism following the KD and HCD had a composition corresponding with the respective diet and the meal consumed after the HD reflected the composition of a typical Western diet [24]. Fasting measures were collected at baseline after each diet condition following a minimum 12-hour overnight fast from food, beverages, supplements and medications, and an abstention from exercise for 24 h; postprandial measures were collected every 30 min after the ingestion of the test meal for three hours.

### 2.3. Diet Conditions

For the KD, participants were asked to consume <10% of their total daily energy from carbohydrates and 15% of their total daily energy from protein with the remaining from dietary fat. During the HCD, participants were required to consume ≥65% of their total daily energy from carbohydrates and <20% of their daily energy from fat 15% coming from protein. Participants were instructed to maintain their normal dietary habits during the HD. All diet interventions were followed ad libitum.

Adherence to the KD and HCD were assessed using digital food logs sent to an investigator (a registered dietitian) daily for feedback and assistance. Overall compliance to the KD and HCD were evaluated using three-day dietary records from the second week of each diet intervention, which were also completed during their HD. Compliance to the KD was verified daily using urinary ketone strips. Compliance was also measured by collecting capillary β-hydroxybutyrate (βHB; KETO-MOJO, Auburn, CA, USA) at day 7 and 14 of the KD. Ketosis was defined as a capillary (0.99 ± 0.25 mmols/L) and urinary (1.82 ± 0.52 mmols/L) βHB level ≥ 0.5 mmol/L. Participants maintained their normal exercise and training habits for the entirety of the study, which were measured and verified using self-reported exercise logs and session rating of perceived exertion (sRPE). Participants monitored and reported their weight daily, and any changes in weight ≥5% were considered exclusionary.

### 2.4. Test Meals

Energy content for each test meal was 60% of the participants RMR measured at baseline. The percentage of energy from protein, carbohydrates, and fat in the ketogenic and high-carbohydrate test meals matched the percentages assigned to the KD and HCD, respectively. The test meal for the HD matched the composition of a typical Western diet [24]. Energy, protein, volume, and fiber content were consistent across all three test meals. A full breakdown of the test meals can be found elsewhere [21].

### 2.5. Metabolic Response

Respiratory gases were measured using indirect calorimetry (TrueOne^®^ 2400, ParvoMedics, Sandy, UT, USA) with a ventilated hood system while the participant lay motionless in the supine position in a climate-controlled room. Respiratory gas measurements were collected at baseline for 30 min to determine fasting EE, and every 30 min for three hours after consumption of the test meal. Respiratory gases were collected for the first 20 min during each 30 min interval, but only 15 min of that 20 min period were used to evaluate metabolic responses [25]. The other 10 min within the 30 min interval were used to collect other measurements described elsewhere [21] and to perform gas calibrations or to provide participants with water (120 mL every hour). EE was calculated using the equations from Mehta et al. [26]:EE (kcal/d) = [(3.941(VCO2/RER) + 1.106(VCO2)] × 1440
and substrate oxidation was calculated using the Frayn equations [27]:FatOx (g/min) = (1.67 × VO2) − (1.67 × VCO2)
CarbOx (g/min) = (4.56 × VCO2) − (3.21 × VO2)

Both assume no protein oxidation.

### 2.6. Procedures

Participants reported to the laboratory for an initial screening visit, which included the completion of demographic, lifestyle, and health history questionnaires, and a V̇O2max test described in detail elsewhere [21].

Upon eligibility, participants reported to the laboratory on a second occasion following a minimum 12-hour overnight fast from food, beverages, supplements and medications, and abstention from exercise to collect descriptive information, including body composition and RMR. Weight was measured using a digital scale and height using a stadiometer. Body composition was measured using air displacement plethysmography (BodPod™, Life Measurement Inc., Concord, CA, USA).

After the first two visits, participants completed three separate experimental visits; one following each diet condition. Participants arrived at the laboratory for each experimental visit after a ≥12-hour overnight fast from food, beverages, supplements and medications, and abstention from exercise for 24 h. Height, weight, capillary βHB, and fasting metabolic measurements were collected upon arrival. Afterward, participants were instructed to consume their assigned test meal within 10 min. Respiratory gases were collected every 30 min for three hours after the ingestion of the test meal. The same procedures were followed at each subsequent trial.

### 2.7. Statistical Analysis

All primary outcome variables for each condition were assessed for normality using Shapiro–Wilk tests and visual inspection of Q-Q plots. One outlier was detected for postprandial CarbOx, but the participant’s fasting CarbOx was normal and their three-hour postprandial time course followed a similar pattern to that of all other participants and, thus, this participant was included. The effect of diet condition and meal condition order were assessed for EE, RER, FatOx, and CarbOx using repeated measures ANOVA and revealed no order effects. Violations of sphericity occurred on one occasion and were corrected using Greenhouse–Geisser.

The chronic effects of diet condition on fasting EE, RER, FatOx, and CarbOx following each two-week dietary intervention were analyzed using repeated measures ANOVA. For fasting EE, unadjusted values and values regressed to the weight measured after each diet intervention were analyzed to account for the effect of weight changes during the KD. Effects of meal condition, time, and interaction on postprandial EE, RER, FatOx, and CarbOx were analyzed using mixed models analysis. Peak postprandial responses were calculated as the highest 30 min value after baseline for EE, FatOx, and CarbOx. Because the ketogenic diet/meal was the condition of primary interest, peak postprandial RER was calculated as the lowest 30 min value after baseline. The incremental area under the curve (iAUC) was calculated for each variable across meal conditions and assessed using repeated measures ANOVA. Post hoc tests were conducted using Bonferroni corrections following each omnibus model. In addition to reporting traditional *p*-values, and to support the interpretation of these data based on contemporary statistical procedures [28,29], effect sizes and their corresponding 95% confidence intervals (95%CI) were reported [30]. Partial eta squared (η^2^_p_) was reported for each omnibus model and Cohens dz was reported following simple effects tests. Mean differences (MD) and their 95%CI were reported when *p*-values for simple effect tests were not significant, but significant effect sizes were observed. Statistical significance was accepted at *p* ≤ 0.05. Statistical analyses were performed using SPSS version 27 and R version 4.1.2 [31].

## 3. Results

Participant characteristics are presented in Table 1. Fasting and postprandial iAUC for EE, RER, FatOx, and CarbOx are presented in Table 2. Postprandial iAUC and the time course for EE, RER, FatOx, and CarbOx are illustrated in Figure 1, Figure 2, Figure 3 and Figure 4. A complete analysis of the control variables between diet conditions is reported in Graybeal et al. [21]. Briefly, there were no differences in energy intake, exercise distance, or sRPE. Changes in body mass were <5.0% across conditions, although body mass was significantly lower following the KD despite having the highest average energy intake among the diet conditions. Measurements of βHB and urinary ketones indicated ketosis and compliance to the KD throughout. Carbohydrate intake was significantly lower and fat significantly higher during the KD (carbohydrate: 8.7 ± 2.9%; fat: 64.1 ± 5.39%) compared to the other diets; carbohydrate intake was significantly higher and fat significantly lower during the HCD (carbohydrate: 63.3 ± 8.8%; fat: 20.8 ± 7.6%) compared to the HD (carbohydrates: 45.8 ± 6.86%; fat: 38.2 ± 7.8%). Percent energy from protein was significantly higher on the KD compared to the other diets (KD: 26.0 ± 2.9%; HCD: 14.4 ± 3.2%; HD:16.5 ± 4.2%).

### 3.1. Energy Expenditure

There were no significant differences in fasting EE by diet condition before (*p* = 0.321; η^2^_p_: 0.20) or after adjusting for the participants weight at the end of each diet (*p* = 0.194; η^2^_p_: 0.31; Table 2). For postprandial EE, there were significant effects of meal condition (*p* = 0.029; η^2^_p_: 0.51) and time (*p* < 0.001; η^2^_p_: 0.79). While not statistically significant (*p* = 0.100), the data were consistent with postprandial EE being lower following the KD_meal_ compared to the HD_meal_ using effect size (d: −1.19, d^95%CI^: −2.23, −0.09) and MD (MD: −3.32, MD^95%CI^: −2.23, −0.09) with no other differences observed. The time course for EE by condition is demonstrated in Figure 1.

There was a main effect of meal condition on postprandial EE_peak_ (*p* = 0.003; η^2^_p_: 0.68); postprandial EE_peak_ was significantly lower following the KD_meal_ compared to the HD_meal_ (*p* = 0.024; d: −1.74, d^95%CI^: −3.04, −0.40). While not statistically significant (*p* = 0.105), the data were consistent with the KD_meal_ having a lower EE_peak_ than the HCD_meal_ based on effect size (d: −1.16, d^95%CI^: −2.19, −0.07) and MD (MD: −2.37, MD^95%CI^: −2.23, −0.09). Postprandial EE_iAUC_ was also significantly different by meal condition (*p* = 0.029; η^2^_p_: 0.51; Table 2 and Figure 1). Although not significantly different (*p* = 0.114), the data were consistent with EE_iAUC_ being lower for the KD_meal_ compared to the HCD_meal_ based on effect size (d: −1.14, d^95%CI^: −2.17, −0.06) and MD (MD: −9.39, MD^95%CI^: −18.00, −0.77).

### 3.2. Respiratory Exchange Ratio

There were no significant effects of diet condition on fasting RER (*p* = 0.147; η^2^_p_: 0.32). For postprandial RER, the effects of meal condition (*p* < 0.001; η^2^_p_: 0.82), time (*p* = 0.001; η^2^_p_: 0.50), and interaction (*p* < 0.001; η^2^_p_: 0.43) were observed. Postprandial RER was significantly lower following the KD_meal_ compared to the HCD_meal_ (*p* < 0.001; d: −4.89, d^95%CI^: −7.93, −1.86). While not statistically significant (*p* = 0.066), our data were consistent with postprandial RER being lower following the KD_meal_ compared to the HD_meal_ based on effect size (d: −1.28, d^95%CI^: −2.37, −0.14) and MD (MD: −0.07, MD^95%CI^: −0.13, −0.01). The time course for RER by condition is illustrated in Figure 2.

Postprandial RER_peak_ was significantly different by meal condition (*p* = 0.002; η^2^_p_: 0.72), where it was lower following the KD_meal_ compared to the HCD_meal_ (*p* = 0.001; d: −3.62, d^95%CI^: −5.92, −1.30) but not the HD_meal_ (*p* = 0.452; d: −0.69, d^95%CI^: −1.57, 0.24). RER_iAUC_ was significantly different between meal conditions (*p* = 0.015; η^2^_p_: 0.57). RER_iAUC_ was significantly lower following the KD_meal_ compared to the HCD_meal_ (*p* = 0.038; d: −1.55, d^95%CI^: −2.75, −0.29) but not the HD_meal_ (*p* = 0.416; d: −0.72, d^95%CI^: −1.60, 0.22).

### 3.3. Fat Oxidation

There were no significant effects of diet condition on fasting FatOx (*p* = 0.132; η^2^_p_: 0.33). For postprandial FatOx, there were significant effects of meal condition (*p* = 0.001; η^2^_p_: 0.77), time (*p* = 0.003; η^2^_p_: 0.47), and interaction (*p* < 0.001; η^2^_p_: 0.49). The effects of meal condition showed that postprandial FatOx was significantly higher following the KD_meal_ compared to the HCD_meal_ (*p* = 0.001; d: 3.38, d^95%CI^: 1.20, 5.55). While not statistically significant (*p* = 0.122), the data were consistent with postprandial FatOx being higher following the KD_meal_ compared to the HD_meal_ based on effect size (d: 1.12, d^95%CI^: 0.04, 2.13) and MD (MD: 0.78, MD^95%CI^: 0.05, −1.52). The time course for FatOx by condition is presented in Figure 3.

Postprandial FatOx_peak_ was significantly different between meal conditions (*p* = 0.002; η^2^_p_: 0.73), where it was significantly higher following the KD_meal_ compared to the HCD_meal_ (*p* = 0.003; d: 2.79, d^95%CI^: 0.92, 4.63) but not the HD_meal_ (*p* = 0.381; d: 0.75, d^95%CI^: −0.20, 1.64). The data were also consistent with FatOx_peak_ being higher for the HD_meal_ compared to the HCD_meal_ based on effect size (d: 1.10, d^95%CI^: 0.03, 2.12) and MD (MD: 1.19, MD^95%CI^: 0.06, 2.31), although not statistically significant (*p* = 0.127). FatOx_iAUC_ was significantly different by meal condition (*p* = 0.001; η^2^_p_: 0.75), showing that FatOx_iAUC_ was significantly higher following the KD_meal_ compared to the HCD_meal_ (*p* = 0.013; d: 2.03, d^95%CI^: 0.55, 3.46), but not the HD_meal_ (*p* = 0.164; d: 1.02, d^95%CI^: −0.02, 2.00). Additionally, FatOx_iAUC_ was significantly higher after the HD_meal_ than the HCD_meal_ (*p* = 0.035; d: 1.58, d^95%CI^: 0.31, 2.79).

### 3.4. Carbohydrate Oxidation

There were no significant effects of diet condition on fasting CarbOx (*p* = 0.101; η^2^_p_: 0.37). For postprandial CarbOx, there were significant effects of meal condition (*p* < 0.001; η^2^_p_: 0.79), time (*p* < 0.001; η^2^_p_: 0.70), and interaction (*p* < 0.001; η^2^_p_: 0.51). Postprandial CarbOx was significantly lower following the KD_meal_ compared to the HD_meal_ (*p* = 0.029; d: −1.66, d^95%CI^: −2.91, −0.35) and HCD_meal_ (*p* = 0.002; d: −3.22, d^95%CI^: −5.30, −1.12). The time course for CarbOx by condition is presented in Figure 4.

Postprandial CarbOx_peak_ was significantly different by meal condition (*p* < 0.001; η^2^_p_: 0.88), where it was significantly lower following the KD_meal_ compared to the HD_meal_ (*p* = 0.006; d: −2.44, d^95%CI^: −4.09, −0.76) and HCD_meal_ (*p* = 0.001; d: −3.78, d^95%CI^: −6.18, −1.38). The data were consistent with CarbOx_peak_ being higher during the HCD_meal_ than the HD_meal_ based on effect size (d: 1.11, d^95%CI^: 0.04, 2.13) and MD (MD: 2.41, MD^95%CI^: 0.14, 4.67), although not significantly different (*p* = 0.124). Postprandial CarbOx_iAUC_ was significantly different by meal condition (*p* = 0.001; η^2^_p_: 0.77), where it was lower after the KD_meal_ compared to the HCD_meal_ (*p* = 0.005; d: −2.47, d^95%CI^: −4.14, −0.77). While not significantly different (*p* = 0.083), the data were consistent with postprandial CarbOx_iAUC_ being lower for the KD_meal_ compared to the HD_meal_ using effect size (d: −1.26, d^95%CI^: −2.33, −0.13) and MD (MD: −14.4, MD^95%CI^: −26.4, −2.37).

## 4. Discussion

This study examined the chronic and postprandial metabolic responses to a KD compared to a HCD and HD in a group of trained competitive cyclists and triathletes. While several studies have reported metabolic responses to a KD in endurance athletes, most studies have focused on exercise specific responses in comparison to a HCD [7,32,33,34,35,36,37]. Our study is unique in its practical approach, where we examined the metabolic responses to an ad libitum KD in comparison to both an ad libitum HCD and the athlete’s HD, which allowed us to compare both the KD and HCD to the athletes current dietary practices and show the effects more likely to be observed outside of a research setting. In addition, endurance athletes following a KD are regularly consuming meals in accordance with the diet’s guidelines and, thus, our study is novel in its examination of postprandial metabolic responses during each diet separate from those observed during exercise. In summary, our study showed that, after a HD and an ad libitum two-week KD and HCD, there were no differences in fasting EE, RER, FatOx, or CarbOx. In addition, we showed that postprandial EE and CarbOx were lower after the ketogenic meal, but that there were no differences in FatOx or peak RER between the KD and HD.

Few studies have examined the effects of a chronic KD on fasting EE in trained endurance athletes [7,32,33,34,35,36,37]. Interestingly, insights toward our findings may be gleaned from studies investigating metabolic responses to a KD in individuals with overweight or obesity. In men with overweight or obesity, Hall et al. [19] previously reported slight increases in EE following a four-week isocaloric KD compared to the participants’ HD. Low active individuals of higher weight status tend to have reduced metabolic flexibility to elevated dietary fat intake, defined as the ability to increase FatOx under the conditions of increased lipid availability, compared to their lean and more active counterparts [38]. Thus, it is possible that an increased dietary fat intake without proportional increases in FatOx would signal increases in EE to maintain energy homeostasis in response to the greater lipid availability. Given that the cyclists and triathletes in our study likely had a greater metabolic flexibility as a result of their training and ideal metabolic health status [39], their greater capacity for lipid oxidation under high-fat conditions may explain the absence of changes in fasting EE following a KD.

On the other hand, a secondary analysis of the study from Hall et al. [40] reported that the increases in EE observed between the HD and KD only occurred on days where participants were free-living outside of a metabolic ward. It is common for physical activity to increase under free-living conditions, resulting in a greater glucose demand. During severe carbohydrate restriction, increased glucose demand from physical activity is met through increased hepatic gluconeogenesis, which may increase EE due to the inefficiency of this metabolic process [40]. Although our study examined cyclists and triathletes in free-living conditions similar to this secondary analysis, EE did not increase during the KD. Because the KD requires the removal of carbohydrates (the primary macronutrient used to fuel training), it is possible that the restrictions resulted in increased feelings of fatigue during training, which manifested in lower non-exercise activity and thus similar net glucose requirements and no increases in EE. While our participants reported no changes in training or sRPE across diets, it is possible that this information did not capture differences in general physical activity, which is ultimately a limitation of our study.

Interestingly, our study showed no differences in fasting substrate oxidation, which is in opposition to similar studies conducted in endurance athletes [32]. As previously stated, endurance trained individuals have a greater capacity to oxidize fat after the initiation of a high-fat diet, particularly in skeletal muscle [41]. However, increases in FatOx may be specific to skeletal muscle in endurance trained individuals, where studies show that improved FatOx from training does not translate to higher 24-hour increases in FatOx [42], similar to our findings. This is an important distinction given that the majority of studies have reported changes in substrate utilization during exercise, which may not explain the chronic whole-day changes in substrate utilization that occur outside of a training environment. It is also noteworthy to highlight the high-fat intake we observed during the HD. Because the cyclists and triathletes in our study were not following a planned dietary strategy during their HD condition, their ad libitum dietary fat intakes were reflective, albeit higher, of a high-fat Western diet. This is unsurprising given that under free-living conditions, recreationally competitive endurance athletes likely have dietary patterns resembling their broader community. Thus, their existing high-fat intake (a trademark component of the KD) may have blunted the metabolic shifts commonly observed during the KD. This may also be why no differences in substrate utilization were observed between the HD and HCD in reference to CarbOx, given the mixed composition of the participants’ current diet.

While fasting EE did not differ between diet conditions, postprandial EE was lower after the KD meal compared to the other conditions. These findings were unsurprising given that the hierarchy of EE following macronutrient ingestion is protein > carbohydrate > fat in sequence from largest to smallest thermogenic response [43]. Considering that energy and protein were controlled across test meals, the higher carbohydrate concentration in the HD and HCD test meals likely resulted in greater postprandial EE compared to the KD. It should be noted, however, that the test meal used to resemble a meal consumed during a KD had a much lower protein content than most KD in practice. This is supported by the 13% increase in energy from protein between the HD and KD in our study, despite regular feedback from a registered dietitian. Thus, it is possible that our KD test meal may not be reflective of a high-protein KD meal in practice, which would, in theory, lead to a higher postprandial EE than the values that we observed. This also assumes adequate energy content in a typical ketogenic meal, which may be difficult in practice given the severe removal of carbohydrates. Future investigations are required to distinguish the differences between a ketogenic meal designed in a laboratory setting and one followed in practice.

The higher fat content in the HD meal may also be why postprandial FatOx did not appear to favor the KD over the HD. Assuming that the HD test meal resembles the typical diet composition for this group of athletes and considering that fasting FatOx between the KD and HD were similar, one could question if there are distinguishable metabolic advantages of a KD compared to the typical Western diet followed by most recreational competitive cyclists and triathletes. Studies have shown that severe carbohydrate restriction impairs exercise economy and other performance measures in endurance athletes [36]. Thus, if an athlete’s HD exhibits fasting and postprandial FatOx similar to a KD but without the caveat of restricting carbohydrates, what is the metabolic advantage of a KD over the athlete’s habitual intake? Certainly, rapid weight loss during a KD may increase speed [44] and power-to-weight ratio (W/kg), but without refined carbohydrate restoration strategies, the likelihood of performance decrements remain. As such, future investigations should examine the effects of these ad libitum diet strategies on sports performance in this population.

Moreover, there are other issues that warrant discussion. Knowing the potential for performance declines, the dramatic shift from the athlete’s typical carbohydrate intake may increase dietary burden without proportional payoffs. The rapid weight loss that may occur from the KD may also foster potential disordered eating behaviors, particularly in female endurance athletes, who are more likely to restrict both energy and carbohydrates [4]. The high satiety value from large proportions of fat and protein during KD may also lead to insufficient daily energy intake relative to exercise load (i.e., LEA). As such, it is possible that KD may contribute to LEA in cyclists and triathletes in addition to the recently observed health issues associated with carbohydrate restriction, independent of LEA [18].

The limitations of this secondary analysis have been reported in prior works [21]. The major limitation of the larger body of work was that the COVID-19 pandemic resulted in a forced termination of the study and, thus, a smaller sample size. However, given the contrast between diet conditions, we were able to observe several significant differences, although significant effect sizes in a smaller sample may not indicate a true effect and should be interpreted in the appropriate context. In addition, we conducted an a priori power analysis for a within-subjects repeated measures design using the lowest effect size of interest observed in our omnibus model (EE, η^2^_p_ = 0.20), an α = 0.05, and a moderate correlation value of 0.50, and showed that an n of six would yield 80% power. Further, we conducted a post hoc power analysis that showed that, using these same criteria, six participants resulted in 84.4% power. Another limitation specific to this analysis was the short diet duration. However, studies have shown that ketosis is reached within three-to-five days on a KD [23], and that metabolic adaptations plateau within a two-week period [19,45]. Higher protein intake during a KD may have influenced fasting EE, but given that protein intake is likely to be higher during a KD in practice, our findings are more representative of the effects likely to be observed in a real-world setting. Although weight loss occurred during the KD, we regressed EE to the weights measured after 14 d on each diet and found no differences. In addition, weight loss during a short-term KD is likely due to losses in body water and glycogen content [46]. Our study also included a large age range that may have influenced our results. However, the within-subjects design accounts for, in part, the between subjects variation in age. Further, cyclists and triathletes in practice spread across wide age ranges with a large proportion of this population being classified as masters athletes [4,47].

## 5. Conclusions

In conclusion, the ad libitum two-week KD did not lead to differences in fasting EE, RER, FatOx, or CarbOx when compared to an ad libitum HCD or HD in trained competitive cyclists and triathletes. Although postprandial EE and CarbOx were lower after the KD meal, postprandial FatOX was not significantly different between the KD and HD; it should be noted that the KD meal had a lower protein content than what may be observed in practice. It appears that, based on our findings, there are few metabolic advantages in regard to FatOx between an athlete’s HD (when the diet pattern is representative of a traditional Western diet) and a KD, where athletes may inherently experience metabolic responses similar to a KD without the caveat of carbohydrate restriction and need for substantial restoration. Cyclists and triathletes should consider these metabolic similarities when attempting a KD given the potential for health and performance impairments during severe carbohydrate restriction. For athletes who choose to employ a KD, a short implementation period away from a high-stakes competition may be warranted to evaluate performance differences from a HD and to practice carbohydrate restoration and timing strategies prior to a competition.

## Figures and Tables

**Figure 1 ijerph-20-01110-f001:**
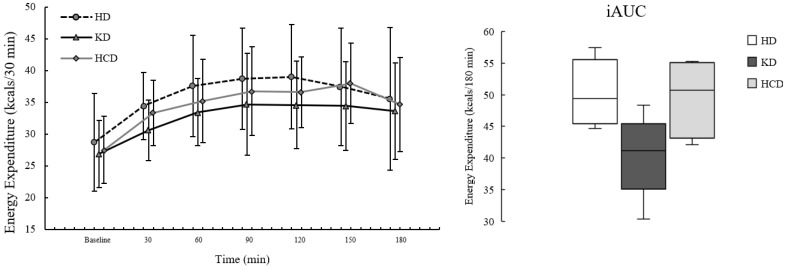
Postprandial energy expenditure (EE) following a ketogenic meal (KD), a high-carbohydrate meal (HCD), and a habitual meal (HD). Each point on the line graph is plotted as mean and standard deviation. The boxplots represent the incremental area under the curve (iAUC) where the line within each box represents the 50th percentile; the upper and lower limits of each box represent the 25th and 75th percentile; and the upper and lower error bars representing the 10th and 90th percentile. There were no significant effects between condition and time or for iAUC (*p* < 0.050). iAUC: incremental area under the curve; HD: habitual diet meal; KD: ketogenic diet meal; HCD: high-carbohydrate meal.

**Figure 2 ijerph-20-01110-f002:**
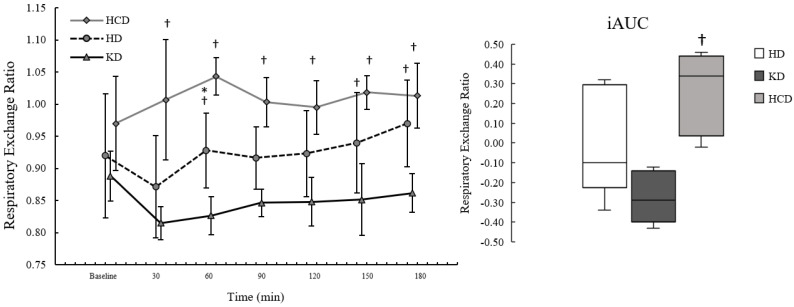
Postprandial respiratory exchange ratio (RER) following a ketogenic meal (KD), a high-carbohydrate meal (HCD), and a habitual meal (HD). Each point on the line graph is plotted as mean and standard deviation. The boxplots represent the incremental area under the curve (iAUC) where the line within each box represents the 50th percentile; the upper and lower limits of each box represent the 25th and 75th percentile; and the upper and lower error bars representing the 10th and 90th percentile. RER was significantly lower at all timepoints following the KD_meal_ compared to the HCD_meal_ (all *p* < 0.050) and at 60 min (*p* = 0.034), 150 min (*p* = 0.038), and 180 min (*p* = 0.017) compared to the HD_meal_. Postprandial RER was significantly higher following the HCD_meal_ compared to the HCD_meal_ at 60 min (*p* = 0.024). RER_iAUC_ was significantly lower following the KD_meal_ compared to the HCD_meal_ (*p* = 0.038). HD: habitual diet meal; KD: ketogenic diet meal; HCD: high-carbohydrate meal. † = significantly different from the KD at *p* ≤ 0.050. * = significantly different from the HCD at *p* ≤ 0.050.

**Figure 3 ijerph-20-01110-f003:**
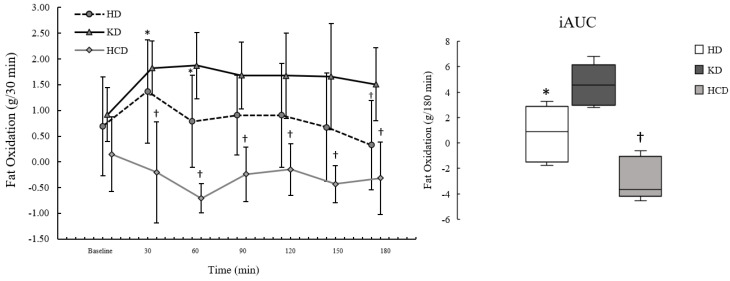
Fat oxidation (FatOx) following a ketogenic meal (KD), a high-carbohydrate meal (HCD), and a habitual meal (HD). Each point on the line graph is plotted as mean and standard deviation. The boxplots represent the incremental area under the curve (iAUC) where the line within each box represents the 50th percentile; the upper and lower limits of each box represent the 25th and 75th percentile; and the upper and lower error bars representing the 10th and 90th percentile. Postprandial FatOx was significantly higher at all timepoints for the KD_meal_ compared to the HCD_meal_ (all *p* < 0.050) but only at 180 min compared to the HD_meal_ (*p* = 0.007). In addition, postprandial FatOx was significantly higher during the HD_meal_ at 30 min (*p* = 0.044) and 60 min (*p* = 0.050) compared to the HCD_meal_. FatO_iAUC_ was significantly higher following the KD_meal_ compared to the HCD_meal_ (*p* = 0.013) and for the HD_meal_ compared to the HCD_meal_ (*p* = 0.035). HD: habitual diet meal; KD: ketogenic diet meal; HCD: high-carbohydrate meal. † = significantly different from the KD at *p* ≤ 0.050. * = significantly different from the HCD at *p* ≤ 0.050.

**Figure 4 ijerph-20-01110-f004:**
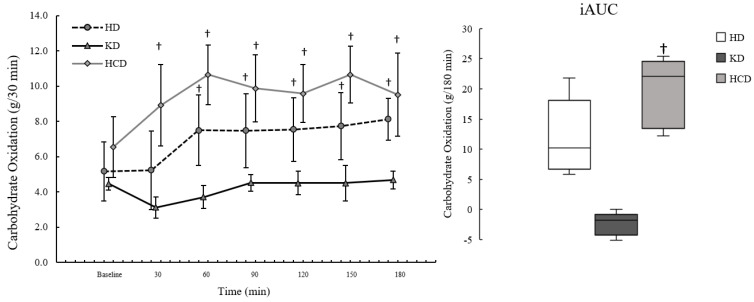
Carbohydrate oxidation (CarbOx) following a ketogenic meal (KD), a high-carbohydrate meal (HCD), and a habitual meal (HD). Each point on the line graph is plotted as mean and standard error. The boxplots represent the incremental area under the curve (iAUC) where the line within each box represents the 50th percentile; the upper and lower limits of each box represent the 25th and 75th percentile; and the upper and lower error bars representing the 10th and 90th percentile. CarbOx was significantly higher at all timepoints following the HCD_meal_ (all *p* < 0.050) and at all timepoints other than 30 min (*p* = 0.298) for the HD_meal_ when compared to the KD_meal_. There were no differences in postprandial CarbOx between the HCD_meal_ and HD_meal_. CarbOx_iAUC_ was significantly lower after the KD_meal_ compared to the HCD_meal_ (*p* = 0.005). HD: habitual diet meal; KD: ketogenic diet meal; HCD: high-carbohydrate meal. † = significantly different from the KD at *p* ≤ 0.050. * = significantly different from the HCD at *p* ≤ 0.050.

**Table 1 ijerph-20-01110-t001:** Participant characteristics.

	Total (n = 6)	Female (n = 4)	Male (n = 2)
Age (y)	37.2 ± 12.2	35.0 ± 9.5	41.5 ± 20.5
Height (cm)	172.3 ± 10.0	167.0 ± 5.5	183.5 ± 0.7
Weight (kg)	68.5 ± 17.5	58.2 ± 8.3	89.1 ± 7.1
BMI (kg/m^2^)	22.7 ± 3.4	20.9 ± 2.0	26.5 ± 2.3
Body Fat (%)	21.3 ± 4.6	21.4 ± 4.2	21.1 ± 7.2
FFM (kg)	53.8 ± 13.2	45.6 ± 5.0	70.1 ± 0.8
FM (kg)	14.7 ± 5.9	12.6 ± 4.2	19.1 ± 7.9
V̇O2max (mL/kg/d)	46.6 ± 6.7	46.3 ± 7.7	47.3 ± 6.7
RMR (kcals/d)	1617.3 ± 314.7	1426.3 ± 132.0	1999.5 ± 68.6

Data are presented as mean ± standard deviation. BMI, body mass index; FFM, fat-free mass; FM, fat mass; RMR, resting metabolic rate.

**Table 2 ijerph-20-01110-t002:** Metabolic responses during fasting and for postprandial area under the curve.

	HD	KD	HCD
**Fasting**			
EE (kcal/30 min)	28.7 (20.6, 36.8)	26.9 (21.4, 32.4)	27.5 (22.0, 33.0)
RER	0.92 (0.82, 1.02)	0.89 (0.85, 0.93)	0.97 (0.89, 1.05)
FatOx (g/30 min)	0.69 (−0.32, 1.70)	0.92 (0.37, 1.47)	0.15 (−0.62, 0.92)
CarbOx (g/30 min)	5.17 (3.41, 6.93)	4.47 (4.09, 4.86)	6.55 (4.74, 8.36)
**iAUC**			
EE (kcal/180 min)	50.5 (44.2, 56.8)	40.0 (29.0, 51.0)	49.4 (42.5, 56.3) ^a^
RER	−0.03 (−0.31, 0.37)	−0.28 (−0.44, −0.13)	0.26 (0.02, 0.50) ^a^
FatOx (g/180 min)	0.81 (−1.69, 3.31)	4.72 (2.64, 6.81)	−2.94 (−4.94, −0.95) ^a^
CarbOx (g/180 min)	12.59 (3.54, 21.7) ^a^	−1.79 (−5.21, −1.62)	19.90 (13.4, 26.4) ^a^

Data are presented as mean (95% confidence interval). N = 6 (F: 4, M:2) for all diet conditions. ^a^ Significantly different than the ketogenic meal at *p* ≤ 0.050. HD, habitual diet; KD, ketogenic diet; HCD, high-carbohydrate diet; EE, energy expenditure; RER, respiratory exchange ratio; FatOx, fat oxidation; CarbOx, carbohydrate oxidation; iAUC, incremental area under the curve.

## Data Availability

Data generated or analyzed during this study are available from the corresponding author upon reasonable request.

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
