# Peer review of "Chronic and Postprandial Metabolic Responses to a Ketogenic Diet Compared to High-Carbohydrate and Habitual Diets in Trained Competitive Cyclists and Triathletes: A Randomized Crossover Trial"

_ijerph, 2023, doi:10.3390/ijerph20021110_

Round 1
Reviewer 1 Report
- Line 19: "KD" Write down the whole word because this is the first time it appears in the abstract.
- Line 88: The author should point out that the metabolic responses recorded were in response to an ad libitum diet.
- Line 147: "Body weight was significantly lower following the KD despite having the highest average energy intake, but the change in body mass but was < 5.0%"... The sentence needs correction and the word "but" is repeated.
- Line 172 & 173: Write down the equations.
- The authors should explain why they prefer to use ad libitum diet rather than a strict dietary program. What is the significance? What are the expected differences in the results?
Author Response
Line 19: "KD" Write down the whole word because this is the first time it appears in the abstract.
Author response: Thank you for catching this. We have included the term before the abbreviation.
Line 88: The author should point out that the metabolic responses recorded were in response to an ad libitum diet.
Author response: Thank you for your suggestion. We have noted the ad libitum distinction in the area of the manuscript that you have recommended.
Line 147: "Body weight was significantly lower following the KD despite having the highest average energy intake, but the change in body mass but was < 5.0%"... The sentence needs correction and the word "but" is repeated.
Author response: Thank you for your suggestion. We have substantially altered this sentence to improve clarity and have deleted the repeated word.
Line 172 & 173: Write down the equations.
Author response: Thank you for your suggestion. We have added the equations to the manuscript as requested.
The authors should explain why they prefer to use ad libitum diet rather than a strict dietary program. What is the significance? What are the expected differences in the results?
Author response: Thank you for your suggestion. We have previously highlighted the significance of this strategy in lines 417-421 and in lines 520-521; specifically noting that our findings are more representative of the results one would expect to see in practice (compared to a laboratory setting).
Reviewer 2 Report
REVIEWER Comments:
Congratulations on choosing a good topic. However, you need to check and correct some parts of the thesis for completeness.
-
Methods:
1. The age range of participants was too large. It would be useful to comment on the limitations of this study.
2. The number of participants per group was too small. Isn't it necessary to conduct a non-parametric test in statistical analysis?
Results:
1. Please indicate the number of participants by group in Table 2.
Discussion:
1. Please explain and write the meaning of LEA again (line 501).
2. It would be good to mention that future research needs to investigate the effect of these diets on sports performance.
Conclusions:
1. Add some more practical applications.
Author Response
Congratulations on choosing a good topic. However, you need to check and correct some parts of the thesis for completeness.
Author response: Thank you for reviewing this manuscript. We hope to adequately address your concerns in the points listed below.
The age range of participants was too large. It would be useful to comment on the limitations of this study.
Author response: Thank you for your comment and suggestion. We have added this to the limitations section of the manuscript (lines 530-534).
The number of participants per group was too small. Isn't it necessary to conduct a non-parametric test in statistical analysis?
Author response: Thank you for your comment and question. Although an argument could be made for non-parametric analyses based on sample size, we chose to use parametric analyses to allow for more rigorous statistical analyses that would not be possible using non-parametric methods. We did not do this without taking initial precautions, however. We conducted an a priori power analysis for a within-subjects repeated measures design using the lowest effect size of interest for the omnibus model (energy expenditure, partial eta squared = 0.20), an α = 0.05, and a moderate correlation value of 0.50, and showed that an n of 6 would yield 80% power. Further, we conducted a post hoc power analysis which showed that using these same criteria, 6 participants resulted in 84.4% power. We have added this information in the limitations section where we discuss our sample size limitations (lines 520-525).
Please indicate the number of participants by group in Table 2.
Author response: Thank you for your suggestion. We have added this information to the Table 2 notes.
Please explain and write the meaning of LEA again (line 501).
Author response: Thank you for your suggestion. We have parenthesized this information to improve clarity.
It would be good to mention that future research needs to investigate the effect of these diets on sports performance.
Author response: Thank you for your suggestion. We have added this information to the discussion section as requested (lines 502-504).
Add some more practical applications.
Author response: Thank you for your suggestion. We have added additional practical applications to the conclusion as requested (lines 550-553).
Reviewer 3 Report
Graybeal et al. conducted a randomized crossover trial titled “Chronic and Postprandial Metabolic Responses to a Ketogenic Diet Compared to a High-Carbohydrate and Habitual Diet in Trained Competitive Cyclists and Triathletes: A Randomized Crossover Trial”. The authors present an interesting manuscript which examined the effects of a ketogenic diet (KD) on metabolic responses in trained endurance athletes, specifically comparing the KD to a high-carbohydrate diet (HCD) and the athletes' habitual diet (HD).
Although the authors present data that surely is of interest for many endurance athletes, the article also has some shortcomings:
- 2.1 Participants
Participants were considered to be trained competitive based on guidelines cited in the article. Reference 22 however, only refers to guidelines to classify female subject groups. Were the same guidelines used to classify male subjects in the present study? If so, please discuss/justify.
- 2.2 Design
The authors state that no defined washout period was used. Does this imply that subjects immediately initiated their second diet after completing the first or were there unstandardized washout periods used between subjects? Please specify.
The authors argue that ketosis is immediately washed out after initiating a HCD. Similarly, and for the sake of completeness, please state the theoretical necessary wash out period for a HCD after switching to a HD/KD. It is important to justify the lack of a washout period.
- 2.3 Diet Conditions
Lines 145-155 should be moved to the results section.
- Results
The results section is long and repetitive of Table 2. I suggest shortening. The authors may consider only discussing significant findings.
- Figures 1-4
Please add units to the X-Axis
It is somewhat hard to link the symbols “†“ and “*“ to their respective curves. Please standardise.
Author Response
Participants were considered to be trained competitive based on guidelines cited in the article. Reference 22 however, only refers to guidelines to classify female subject groups. Were the same guidelines used to classify male subjects in the present study? If so, please discuss/justify.
Author response: Thank you for your comment and question. The same guidelines were used for both male and female participants. The reference used to support this evaluates female athlete guidelines against male athlete guidelines and shows that there are sex differences in relative VO2max between classifications. As such, our VO2max classifications were sex-and-age specific to account for the higher VO2max per classification in males.
The authors state that no defined washout period was used. Does this imply that subjects immediately initiated their second diet after completing the first or were there unstandardized washout periods used between subjects? Please specify. The authors argue that ketosis is immediately washed out after initiating a HCD. Similarly, and for the sake of completeness, please state the theoretical necessary wash out period for a HCD after switching to a HD/KD. It is important to justify the lack of a washout period.
Author response: Thank you for your comments and question. We immediately switched participants over to the other diet after completion of the first diet and have added this to the manuscript (line 115). Additionally, we have further justified the lack of a washout period in the opposing direction (HCD to KD) using data from our study (lines 118-121).
Lines 145-155 should be moved to the results section.
Author response: Thank you for your suggestion. We have moved this information to the first paragraph of the results section as requested.
The results section is long and repetitive of Table 2. I suggest shortening. The authors may consider only discussing significant findings.
Author response: Thank you for your suggestion. We have removed information regarding insignificant findings unless there were integral to the interpretation.
Please add units to the X-Axis
Author response: Thank you for catching this. We have added units to the x-axis.
It is somewhat hard to link the symbols “†“ and “*“ to their respective curves. Please standardise.
Author response: Thank you for your suggestion. We have standardized Figure 3 which was unstandardized relative to the other figures in the manuscript.